# Gut Microbiota and Fear Processing in Women Affected by Obesity: An Exploratory Pilot Study

**DOI:** 10.3390/nu14183788

**Published:** 2022-09-14

**Authors:** Federica Scarpina, Silvia Turroni, Sara Mambrini, Monica Barone, Stefania Cattaldo, Stefania Mai, Elisa Prina, Ilaria Bastoni, Simone Cappelli, Gianluca Castelnuovo, Patrizia Brigidi, Massimo Scacchi, Alessandro Mauro

**Affiliations:** 1“Rita Levi Montalcini” Department of Neurosciences, University of Turin, Via Cherasco 15, 10126 Turin, Italy; 2I.R.C.C.S. Istituto Auxologico Italiano, U.O. di Neurologia e Neuroriabilitazione, Ospedale San Giuseppe, Str. L Cadorna 90, 28824 Piancavallo, Italy; 3Unit of Microbiome Science and Biotechnology, Department of Pharmacy and Biotechnology, University of Bologna, Via Belmeloro 6, 40126 Bologna, Italy; 4I.R.C.C.S. Istituto Auxologico Italiano, Laboratorio di Ricerca Metabolica, Ospedale San Giuseppe, Str. L Cadorna 90, 28824 Piancavallo, Italy; 5International Center for the Assessment of Nutritional Status (ICANS), Department of Food, Environmental and Nutritional Sciences (DeFENS), University of Milan, Via G. Celoria 2, 20133 Milan, Italy; 6Department of Medical and Surgical Sciences, Unit of Microbiomics, University of Bologna, Via Massarenti 11, 40126 Bologna, Italy; 7I.R.C.C.S. Istituto Auxologico Italiano, Laboratorio di Neurobiologia Clinica, Ospedale San Giuseppe, Str. L Cadorna 90, 28824 Piancavallo, Italy; 8I.R.C.C.S. Istituto Auxologico Italiano, Laboratorio di Psicologia, Ospedale San Giuseppe, Str. L Cadorna 90, 28824 Piancavallo, Italy; 9Department of Psychology, Catholic University of Milan, Largo Agostino Gemelli 1, 20123 Milan, Italy; 10I.R.C.C.S. Istituto Auxologico Italiano, Divisione di Medicina Generale ad indirizzo Endocrino-Metabolico, Ospedale San Giuseppe, Str. L Cadorna 90, 28824 Piancavallo, Italy; 11Department of Clinical Sciences and Community Health, University of Milan, Via Festa del Perdono 7, 20122 Milan, Italy

**Keywords:** obesity, gut microbiota, dysbiosis, facial emotion recognition, fear, temperament

## Abstract

The microbiota–gut–brain axis extends beyond visceral perception, influencing higher-order brain structures, and ultimately psychological functions, such as fear processing. In this exploratory pilot study, we attempted to provide novel experimental evidence of a relationship between gut microbiota composition and diversity, and fear-processing in obesity, through a behavioral approach. Women affected by obesity were enrolled and profiled for gut microbiota, through 16S rRNA amplicon sequencing. Moreover, we tested their ability to recognize facial fearful expressions through an implicit-facial-emotion-recognition task. Finally, a traditional self-report questionnaire was used to assess their temperamental traits. The participants exhibited an unbalanced gut microbiota profile, along with impaired recognition of fearful expressions. Interestingly, dysbiosis was more severe in those participants with altered behavioral performance, with a decrease in typically health-associated microbes, and an increase in the potential pathobiont, Collinsella. Moreover, Collinsella was related to a lower expression of the persistence temperamental trait, while a higher expression of the harm-avoidance temperament, related to fear-driven anxiety symptoms, was linked to Lactobacillus. Once confirmed, our findings could pave the way for the design of innovative microbiome-based strategies for the treatment of psychological and emotional difficulties by mitigating obesity-related consequences and behaviors.

## 1. Introduction

As a primary emotion, fear is a motivational state, aroused by specific threatening stimuli, manifested in defensive behavior or escape, to avoid or reduce potential harm [1]. The amygdala—together with the afferent hippocampus, medial prefrontal cortex, and efferent periaqueductal gray—constitutes a key brain structure in fear circuitry [2,3,4], rapidly processing fear-related stimuli out of the individual’s consciousness [5]. In human experience, fear represents the emotional underpinning of anxiety-related behaviors [6,7]. 

Evidence from animal models suggests that amygdala-dependent, fear-related behaviors are linked to gut microbiota manipulation [8,9,10,11]. On the other hand, experimental evidence on the role of gut microbiota in processing fear in humans is still in its infancy. Gut microbes are known to potentially influence psychological functions across a lifespan, through multiple gut–brain interactions [12,13,14], such as temperament (i.e., the stable, biologically based component of the individual personality, which shapes behavior, affectivity, and decision-making processes) [15] or reaction to distress [16]. However, from a methodological point of view, most of the evidence in the field has been obtained through self-report descriptions (i.e., questionnaires), enhancing possible implicit response bias. Only very rare evidence is available about the human responses to fearful stimuli explored through an experimental behavioral approach. It is worth mentioning the recent study by Aatsinki and colleagues [17], who explored the associations between fecal microbiota composition and attention to emotional faces—specifically happy and fearful expressions—in infants. The authors observed the association of a lower abundance of Bifidobacterium and a higher abundance of Clostridium with an increased ‘fear bias’, meaning a higher level of attention towards fearful expressions. However, this very interesting evidence could not be extended to adults. 

Here, we aimed to provide the first evidence of a relationship between fear-related mechanisms and human gut microbiota, through an experimental approach, in adults with obesity. We focused on obesity as, from a psychological point of view, affected individuals show higher anxiety-related symptoms [18,19], thus representing a clinical model to investigate the relationship between fear-related mechanisms, in which anxiety is grounded [6,7], and gut microbiota. An altered sensitivity to fear-related facial stimuli in obesity has recently been recognized [20,21]. On the other hand, alterations in brain activity in the amygdala [22], and alteration of gray matter volume in the amygdala, thalamus, and putamen [23,24] (all involved in fear processing), have been observed as possible side effects of chronic low-grade inflammation on cerebral anatomical and functional mechanisms [25]. Overall, all previous (albeit rare) behavioral and neurophysiological evidence may point towards an impaired fear processing in obesity. Interestingly, from a psychological point of view, harm-avoidance temperament—which is related to behavioral inhibition, and is associated with increased serotonergic activity [15], as well as a tendency to lose control over food intake and weight gain [26]—has been extensively described in obesity [27,28,29]. Therefore, this temperamental trait correlates with increased stress perception, leading to activation of the hypothalamic–pituitary–adrenal axis. Not least, typical features of obesity are gut microbiota imbalances (i.e., dysbiosis), with loss of diversity, and reduced proportions of health-associated microorganisms, including short-chain fatty acid (SCFA) producers and the mucus degrader Akkermansia, in addition to increased amounts of pathobionts (e.g., Collinsella) [30,31,32,33]. Gut microbiota may influence the host metabolism, and therefore the onset and progression of obesity, through several pathways, which involve reduced gut barrier integrity and metabolic endotoxemia, the production of metabolites that affect satiety, insulin resistance and lipolysis, epigenetic factors, and bile acid metabolism [30,31,32]. 

In this exploratory pilot study, we investigated the sensitivity of individuals with obesity to fear-related stimuli, through a well-established psychological paradigm—namely, the implicit-facial-emotion-recognition task [20,34]. Through this task, the behavioral detection and recognition of fearful expressions are tested according to the psychophysical phenomenon known as the redundant target effect [35] applied to emotional stimuli [36,37]. Furthermore, we evaluated the expression of our participants’ temperamental traits within the framework of the Cloninger’s model [15]. Gut microbiota was profiled by 16S rRNA amplicon sequencing of fecal samples, and we explored its relationship with individual temperament and behavior, in the course of fear-related tasks. 

Importantly, in this study, we gathered a wealth of information related to psychological wellbeing, diet, and biochemical variables known to be associated with gut microbiota composition [38]. Moreover, we only tested females (and not males) affected by obesity. This methodological decision was taken on account of various considerations. Firstly, gender-specific differences in body morphology, particularly fat-distribution in obesity, are well recognized [39,40,41]. Moreover, gut microbiota has been reported to vary by gender, and estrogen and androgen metabolisms appear to be related to gut microbiota profiles (for a recent review, [42]). Crucially, for the topic of this pilot study, females and males differ in emotional experiences and expressions [43,44]. Finally, we only enrolled women with obesity, in order to be able to compare the experimental data shown in this work with the evidence reported in our previous study [20], in which we tested a group of women, affected by obesity, with the same emotional task. 

## 2. Materials and Methods

### 2.1. Study Design, Enrollment and Questionnaires

This observational, exploratory pilot study was conducted with a quasi-experimental design. It was approved by the I.R.C.C.S. Istituto Auxologico Italiano Ethics Committee (ID: 21C126_2021), following the guidelines of the European Convention on Human Rights and Biomedicine. The subjects volunteered to participate, and gave informed written consent, being free to withdraw at will, and unaware of the rationale of the experiment.

The participants were consecutively recruited, on admission to the I.R.C.C.S. Istituto Auxologico Italiano, Divisione di Endocrinologia e Malattie Metaboliche, Ospedale San Giuseppe, Piancavallo, Italy. Participants were recruited prior to an intensive hospital-based and medically managed rehabilitation program for weight reduction. The recruitment was carried out in 2020, between February and September, with a hiatus between April and May due to the global COVID-19 pandemic. No individual hospitalized for SARS-Cov-2 infection was recruited, given the results reported by Scarpina and colleagues [45]. Given the gender-driven differences in psychological processing, obesity, and microbiota mentioned above (see end of Introduction), we only enrolled women. Only right-handed participants were tested, as a motor response was required in the experimental task. Participants with a body mass index (BMI) greater than 30 kg/m^2^, as an index of obesity [46], were included. As in Cancello et al. [47], the exclusion criteria were: (1) history of colorectal surgery or bariatric surgery; (2) known or suspected inflammatory bowel disease and/or proctitis; (3) prebiotic, probiotic, and/or vitamin supplementation in the previous three weeks; (4) therapy with antibiotics, proton pump inhibitors, and/or metformin; and (5) previous or concomitant history of cancer. Moreover, we excluded individuals with: (6) a history of alcohol abuse; and (7) a history of gastrointestinal, cardiovascular, psychiatric, neurological disorders, or any concurrent medical condition unrelated to obesity, according to the clinical definition of ‘Metabolically Healthy Obesity’ (for a review, see [48]).

Participants’ eating habits were collected by expert dieticians through the Medi-lite questionnaire [49], consisting of nine questions investigating the frequency of daily or weekly consumption of typical (i.e., fruit, vegetables, cereals, legumes, and fish) and non-typical (i.e., meat, meat products, and dairy products) foods of the Mediterranean diet. For typical foods, 2 points were assigned for the highest frequency, 1 for the intermediate frequency, and 0 for the lowest frequency. For non-typical foods, a score of 2 was given for the lowest consumption, 1 for the intermediate consumption, and 0 for the highest consumption. The final scoring could range from 0 (low adherence) to 18 (high adherence).

The Italian version [50] of the Psychological General Wellbeing Index (PGWBI) [51]) was used to describe the subjective perception of wellbeing, according to six dimensions: anxiety; depressed mood; positive wellbeing; self-control; general health; and vitality. The questionnaire consisted of 22 self-administered items, rated on a 6-point Likert scale, in each of the six domains. Each score (0 to 5) referred to the subject’s last four weeks of daily life. Each domain was defined by a minimum of three to a maximum of five items. The scores for all domains were summarized into a global summary score, which reached a theoretical maximum of 110 points, representing the best attainable level of wellbeing. 

### 2.2. Implicit-Facial-Emotion-Recognition Task

The implicit-facial-emotion-recognition task, described by Scarpina and colleagues [20,34], was used. The task enables the measurement of an individual’s ability to implicitly recognize the expression of fear in others’ faces. It is grounded in the redundant target effect [35,36,37], which expresses the general effect of facilitation in cases of redundant targets compared to unique ones. As the attentional phenomenon occurs at a very early level of visual processing, it is not biased by decisional or premotor mechanisms [35,52,53]. This effect has been successfully observed, not only in cases of visual neutral stimuli, but also for emotional faces [36,37]. In accordance with the relevant literature ([35,36,37,54] (and the results provided by our group [20,34]), people respond faster (i.e., with a shorter reaction time) and more accurately when two identical targets are presented simultaneously, or when the target is presented alone, when compared with experimental conditions in which a non-identical stimulus (i.e., the distractor, here represented by a neutral or another emotional face) is presented along with the target. The competitive presence of a non-identical stimulus (i.e., another emotion) affects the efficient recognition of the target, with a slower speed in stimulus detection, and a reduction in the level of accuracy. 

Photographs of male and female faces from a traditional database were used [55] as visual stimuli. A fearful expression was presented in four different conditions: (1) unilateral, with the target (fear) presented on the right OR left of a fixation cross; (2) bilateral congruent, with the target presented simultaneously on the right AND left of the fixation cross; (3) bilateral incongruent emotional, with the target presented on the right OR left of the fixation cross, along with a different emotion (such as an angry face); and (4) bilateral incongruent neutral, with the target presented on the right OR left of the fixation cross, along with a neutral expression. Catch trials were also used, with a distractor (represented by an emotion other than fear) presented unilaterally, bilaterally, or in opposition to a neutral and another emotional stimulus. 

The participants were asked to press a button on a keyword as soon as they recognized a fearful face on the laptop screen. The stimuli stayed on the screen until the participants responded, or for a duration of 1500 milliseconds (ms). The inter-stimulus interval varied randomly between 650 and 950 ms. For each condition (unilateral, bilateral congruent, bilateral incongruent emotional, and bilateral incongruent neutral), 32 valid trials and 16 catch trials were presented. Overall, 384 trials were administered, split into two experimental blocks. 

Two behavioral parameters were recorded (i.e., when the emotion was correctly detected): the Reaction Time (RT) in ms from the stimulus onset, with respect to the valid trials, corresponding to the detection of the target; and the Level of Accuracy (calculated as ‘number of hits—number of false alarms’, and expressed as a percentage), corresponding to the recognition of the target.

### 2.3. Temperament

The Temperament and Character Inventory—revised version by Martinotti and colleagues [56]—was used to assess the four main temperamental traits, according to the Cloninger’s model [15], each of which was associated with an emotion or feeling: (i) novelty-seeking, associated with the emotion of anger, which expressed the level of activation of the exploratory activity; (ii) harm avoidance, related to fear, which reflected the efficiency of the behavioral inhibition system; (iii) reward-dependence, associated with the attachment style, which referred to reward-based behavioral maintenance; and (iv) persistence, related to ambition, which expressed the maintenance of behaviors, such as resistance to frustration. The participants filled out the questionnaire immediately after the completion of the implicit-facial-emotion-recognition task. 

### 2.4. Sample Collection

A fecal sample, for gut microbiota analysis, was collected the day after the experimental task. The samples were delivered to the Department of Pharmacy and Biotechnology (University of Bologna, Bologna, Italy) and stored at −80 °C until processing. Another fecal sample was collected for measurement of calprotectin as an index of gut inflammation [57]. In parallel, blood samples were drawn in the morning, under fasting conditions, for hormone measurements (serotonin, adrenaline, noradrenaline dopamine, and cortisol). The serum was separated by centrifugation at 4200 rpm for 10 min at 4 °C, aliquoted, and stored at −80 °C until analysis at the I.R.C.C.S. Istituto Auxologico Italiano, Laboratorio di Neurobiologia Clinica, Piancavallo, Italy. 

### 2.5. Gut Microbiota Profiling by 16S rRNA Amplicon Sequencing

Microbial DNA was extracted from fecal samples, using the ‘repeated bead-beating plus column’ method [58]. In short, each sample was subjected to three successive phases of lysis: (1) chemical, by resuspension in 500 mM NaCl, 50 mM Tris-HCl pH 8, 50 mM EDTA, and 4% SDS; (2) mechanical, using a FastPrep homogenizer (MP Biomedicals, Irvine, CA) in the presence of four 3-mm glass beads and 0.5 g of 0.1-mm zirconia beads (BioSpec Products, Bartlesville, OK, USA); and (3) thermal, by incubation at 95 °C for 15 min. After centrifugation at 13,000 rpm for 5 min, the nucleic acids were precipitated by the sequential addition of 260 μL of 10 M ammonium acetate, one volume of isopropanol, and 70% ethanol. The nucleic acid pellet was suspended in 100 µL of TE buffer, and incubated with 2 μL of DNase-free RNase (10 mg/mL) at 37 °C for 15 min. The subsequent steps, of protein removal and DNA purification, were performed using the DNeasy Blood and Tissue Kit (QIAGEN, Hilden, Germany). For library preparation, the V3-V4 hypervariable regions of the 16S rRNA gene were amplified, using primers 341F and 785R, including overhang adapter sequences for Illumina sequencing, as per the manufacturer’s instructions (Illumina, San Diego, CA, USA). The amplicons were purified using magnetic beads (Agencourt AMPure XP; Beckman Coulter, Brea, CA, USA), and a limited-cycle PCR was performed using Nextera technology. Indexed and purified libraries were pooled at equimolar concentration, denatured, and diluted to 5 pM, prior to sequencing on an Illumina MiSeq platform, following a 2 × 250 bp paired-end protocol. Raw sequencing reads were deposited in the National Center for Biotechnology Information Sequence Read Archive (BioProject ID: PRJNA837468).

The sequences were processed using PANDASeq [59] and QIIME 2 [60]. Quality-filtered reads were binned into amplicon sequence variants (ASVs), using DADA2 [61]. The taxonomic assignment was performed using the VSEARCH algorithm [62] and Greengenes as the reference database. Alpha diversity was calculated using the inverse Simpson index and Shannon index. For beta diversity, the Bray–Curtis dissimilarity was used, to construct Principal Coordinates Analysis (PCoA) graphs. 

### 2.6. Biochemical Parameters

Serotonin and catecholamine levels in serum were measured, using commercially available ELISA kits (IBL International, Hamburg, Germany). The intra-assay coefficients of variation were 3.8–6.6% for serotonin, 6.8% for adrenaline, 7.4% for noradrenaline, and 10.9% for dopamine. The inter-assay coefficients of variation were 6.7–17.3% for serotonin, 15.2% for adrenaline, 12.5% for noradrenaline, and 16.3% for dopamine. All samples were assayed in duplicate, in accordance with the manufacturer’s instructions. Fecal levels of calprotectin were measured using the Quantum Blue^®^Calprotectin High Range Test (BÜHLMANN Laboratories AG, Schönenbuch, Switzerland), considering the normal range as <100 μg/g. Cortisol, the hormone released during threatening conditions as a fight-or-flight response—thereby representing a measure of a stressful condition—was measured in the blood, using a chemiluminescent assay (Roche Diagnostics, Mannheim, Germany), considering a normal range of 4.82–19.5 μg/dL. 

### 2.7. Statistical Analysis

#### 2.7.1. Experimental Task

As done in previous studies [20,34], reaction time in ms, from the stimulus onset for the valid trials, and the percentage of accuracy were computed for each of the four experimental conditions. To verify the redundant target effect, the two parameters were analyzed independently by means of a repeated-measure ANOVA, with the within-subjects factor of Condition (four levels: unilateral, bilateral congruent, bilateral incongruent emotional, and bilateral incongruent neutral). Estimated marginal mean comparisons with Bonferroni correction were applied in the case of a significant main effect. For the subsequent analyses, an index was calculated for the reaction time and accuracy parameters as: mean (bilateral incongruent emotional and bilateral incongruent neutral)/mean (unilateral and bilateral congruent). Values higher than 1 on the reaction time, and less than 1 on the accuracy level, suggested a behavioral performance in line with the redundant target effect. For both indexes, the median value in the whole group was calculated. Then, participants were stratified into two groups, considering higher and lower values than the median.

#### 2.7.2. Temperament

To describe our sample according to the four temperamental traits, raw scores for each trait of The Temperament and Character Inventory were compared with the Italian normative data provided by Martinotti and colleagues [56], through an independent sample t-test. For subsequent analyses, the raw scores were converted to T-scores (distribution’s M = 50; SD = 10), stratifying the subjects into three groups: upper, lower, and in the medium normative range [56]. 

#### 2.7.3. Gut Microbiota Profiling and Correlation with Host Metadata

All statistical analyses were performed using the R software. To verify the presence of dysbiosis in our participants, alpha diversity and composition at different phylogenetic levels were compared, through the Wilcoxon test, with those of age-matched, normal-weight, healthy Italian women (N = 19; age 41.3 ± 6.3 years; BMI = 21.0 ± 1.6 kg/m^2^) (unpublished data by Barone and colleagues, and [63]; MG-RAST mgp12183). We purposely selected subjects matched, as accurately as possible, to patients for major microbiota-associated confounding factors (i.e., age, gender, and geography) [38], and whose samples had undergone the same wet procedures (i.e., they were processed in the same lab), to avoid study-related bias. It should be noted that the inclusion of published data from previous studies for comparative purposes is a common practice in microbiota studies (e.g., [64]). For beta diversity, the significance of the separation between study subjects and controls, in the Bray–Curtis-based PCoA, was tested using a PERMANOVA (adonis function in R vegan package) or ANOSIM test. 

We then explored the associations between gut microbiota and experimental task parameters (reaction time and level of accuracy), and temperamental traits (novelty-seeking, harm avoidance, reward dependence, persistence) as continuous or categorical variables in our cohort. Specifically, we followed two approaches. Firstly, we sought for univariate correlations between the relative abundances of genera and the values of reaction time and accuracy level for each experimental condition (bilateral incongruent neutral, bilateral incongruent emotional condition, bilateral congruent, and unilateral), using the Kendall rank correlation test; similarly, we verified such a correlation for the T-scores for each temperamental trait. Only statistically significant correlations with absolute Kendall’s tau ≥0.3, and bacterial genera with a relative abundance ≥1%, were considered. Secondly, for each behavioral parameter/temperamental trait, we considered the stratification of our participants (i.e., by higher and lower value than the median for behavioral parameters; and by high, medium, or low value for temperamental traits), and verified whether there were differences in microbiota diversity and composition between these groups and the controls by PERMANOVA, ANOSIM, and Wilcoxon test. Associations between genus-level relative abundances and BMI and biochemical parameters were also sought, through the Kendall rank correlation test. By using the false discovery rate (FDR) method, p-values were corrected for multiple comparisons. A corrected p-value ≤ 0.05 was considered statistically significant, while a p-value between 0.05 and 0.1 was considered a trend. 

#### 2.7.4. Sample Size

No statistical methods were used to predetermine sample size, but we performed a convenience sampling of 20 participants. As no previous studies were available in the field, we used a conservative approach, adopting a small effect size (0.25). Considering an α value of 0.05, the power was 0.29 [65].

## 3. Results

### 3.1. Participants

Twenty women affected by obesity were enrolled (age in years: mean = 43.35, SD = 6.51, range = 27–54; education in years: mean = 14, SD = 3.61, range = 8–18; BMI in kg/m^2^: mean = 49.14, SD = 8.32, range = 39.03–69.18). However, as one fecal sample was missing, only 19 participants were included in the final dataset. The clinical, psychological, and biochemical characteristics of the tested participants are reported in Table 1. 

The results related to the Medi-lite questionnaire [49], which was used to describe our participants’ eating habits, are shown in Table 2. All participants had a diet characterized by a lack of regular and balanced meals in macronutrients. Specifically, an excessive consumption of cheese and processed meats was observed, along with a low consumption of fruit, vegetables, and wholemeal foods, with a consequent low intake of fiber and a high intake of sodium. Excessive consumption of baked food, both sweet and salty, was also observed. In summary, the participants showed a dysfunctional diet, characterized by continuous piling up of food during the day. The average final score at the Medi-lite questionnaire was 9.07 (SD = 2.36, range = 5–12), indicating, as expected, a low adherence to the Mediterranean diet [49]. 

Regarding psychological wellbeing (tested through PGWBI; [51]), we verified whether the scores reported by our participants at each subscale fell within the normative range relative to the Italian population [50] through an independent sample t-test. They reported a lower level of psychological wellbeing (total score: t = 3.62; p = 0.001). Specifically, lower scores were observed for positive wellbeing (t = 2.59; p = 0.01), general health (t = 3.72; p = 0.001), and vitality (t = 4.25; p = 0.004), and a trend was observed for self-control (t = 2.02; p = 0.057). There were no differences in scores related to anxiety (t = 1.14; p = 0.17) and depression (t = 0.8; p = 0.39); these results were crucial for our study, as the level of anxiety and depressive symptoms can impact the facial-emotion-recognition task [66,67]. 

### 3.2. Implicit-Facial-Emotion-Recognition Task

Overall, 22.43% of false alarms were registered in our dataset. Out of two standard deviations, 3.41% of the total trials were removed from the valid ones. Preliminary inspection of the raw data showed the presence of outliers. However, in line with previous studies [20,34], the outliers were not removed. 

Regarding the reaction time (i.e., detection), the main effect of *Condition* was significant [F(3.54) = 5.53; *p* = 0.002; η_p_^2^ = 0.23]. Our participants reported a faster reaction time in the bilateral congruent condition than in the bilateral incongruent neutral condition (*p* = 0.028) and the bilateral incongruent emotional condition (*p* = 0.007); all other comparisons were not significant (*p* ≥ 0.48) (Figure 1, upper panel). Behavioral performance was in agreement with previous evidence described by Scarpina and colleagues [20], relating to a sample of individuals with obesity. This pattern suggested the absence of the *redundant target effect*, which is an altered detection of fearful expressions. 

As regards the level of accuracy (i.e., recognition), the main effect of *Condition* was significant [F(3.54) = 24.83; *p* < 0.001; η_p_^2^ = 0.58]. The participants reported the same level of accuracy in the bilateral incongruent neutral condition and the bilateral incongruent emotional condition (*p* = 1), as well as in the bilateral congruent and the unilateral condition (*p* = 0.25). All other comparisons were significant (*p* ≤ 0.001). Therefore, when the recognition level of fearful expressions was analyzed, the *redundant target effect* clearly emerged (Figure 1, lower panel). This pattern was not in agreement with the previous evidence [20] of a very altered performance in a sample of individuals affected by obesity, when compared to healthy controls. 

### 3.3. Temperament

In line with previous evidence [27,28,29], our participants showed higher expression of harm avoidance and reward dependence, in comparison to the Italian normative data [56]. No differences emerged, in regard to the levels of novelty-seeking and persistence (Table 3).

### 3.4. Gut Microbiota

Sequencing of the 16S rRNA gene of fecal samples, from our sample of women with obesity, yielded a total of 750,189 reads (39,484 ± 8219 per sample), binned into 1667 ASVs. 

Gut microbiota profiles were compared with those of age-matched, normal-weight Italian women as controls (Barone et al. unpublished data; [63]) (Figure 2). Consistent with previous evidence [47,68,69,70,71], alpha diversity was lower in participants than in controls (*p* ≤ 0.04, Wilcoxon test) (Figure 3A). Furthermore, participants segregated from controls in the Bray–Curtis-based PCoA (*p* = 0.0005, PERMANOVA) (Figure 3B). At the compositional level, our participants with obesity were predictably enriched in *Collinsella* [70,71], while depleted in a range of beneficial microorganisms, including major SCFA producers (i.e., *Ruminococcaceae* members, *Lachnospira*, and *Coprococcus*) and other genera known to be related to leanness and metabolic health, such as *Christensenellaceae* [72] and *Akkermansia* [73] (*p* ≤ 0.02, Wilcoxon test; Figure 3C).

### 3.5. Relationship between Gut Microbiota and Experimental Task

Among our participants, no significant correlation was found between gut microbiota composition and reaction time. On the other hand, the relative abundance of *Clostridium* correlated positively with the percentage of accuracy in each experimental condition (tau ≥ 0.504, *p* ≤ 0.007, Kendall rank correlation test) (Figure 4A). When considering cohort stratification (Figure 5), there was no difference between the three groups (i.e., high value, low value, and controls) in alpha and beta diversity, except for a trend towards segregation between participants with high and low reaction time in the Bray–Curtis-based PCoA (*p* = 0.059, ANOSIM). Moreover, participants with low values of reaction time and levels of accuracy showed less alpha diversity than the controls (*p* ≤ 0.02, Wilcoxon test). Compositionally, those participants with low reaction time (i.e., a faster response) exhibited lower relative abundances of unclassified genera of the *Ruminococcaceae* family than those with high reaction time (*p* = 0.02), and tended to show higher proportions of *Megamonas* (*p* = 0.07) and *Prevotella* (*p* = 0.1). On the other hand, those with higher reaction time (i.e., a slower response) were distinguished by lower amounts of *Lachnospira*, *Coprococcus*, and *Akkermansia*, while higher amounts of *Collinsella* and other *Coriobacteriaceae* taxa, as well as *Sutterella*, compared to the controls (*p* ≤ 0.05). As for accuracy, participants with high percentage values (i.e., a better performance) showed higher amounts of *Erysipelotrichaceae* than those with low percentage values (*p* = 0.01). On the other hand, participants with low levels of accuracy (i.e., a worse performance) were characterized by higher amounts of *Collinsella* and other *Coriobacteriaceae* members, and lower amounts of *Parabacteroides* than the controls (*p* ≤ 0.03).

### 3.6. Relationship between Gut Microbiota and Temperament

Among our participants, the expression of the novelty-seeking trait correlated positively with the relative abundance of *Roseburia* (tau = 0.73, *p* = 6 × 10^−5^, Kendall rank correlation test), and the persistence trait with that of *Sutterella* (tau = 0.533, *p* = 0.004) (Figure 4B). Considering the stratification of the cohort by high, medium or low expression of each temperamental trait (Figure 6), diversity values were, or tended to be, lower in those participants with low-to-medium expression for all temperamental traits, compared to the controls (*p* ≤ 0.06, Wilcoxon test). As for the persistence trait, participants with low expression tended to show less alpha diversity even than those with medium expression (*p* = 0.09). In the PCoA of beta diversity, participants with medium *versus* low persistence expression and high vs medium reward-dependence expression tended to segregate (*p* ≤ 0.09, PERMANOVA). From a compositional standpoint, participants with low persistence expression manifested an increased relative abundance of *Coriobacteriaceae* (i.e., *Collinsella*), along with a depletion of *Christensenellaceae* and *Lachnospira*, compared to the controls (*p* ≤ 0.02, Wilcoxon test). Interestingly, these differences were (*Lachnospira*, *p* = 0.003), or tended to be (*Collinsella*, *p* = 0.1), significant even when compared to participants with a medium level of expression. As for reward dependence, while sharing a high representation of *Coriobacteriaceae* compared to the controls (*p* ≤ 0.03), participants with medium and high expression of this temperamental trait differed in the proportions of *Christensenellaceae* (*p* = 0.09), *Lachnospiraceae* members (*p* ≤ 0.09), *Ruminococcus* (*p* = 0.03), and *Dialister* (*p* = 0.03), all of which were (or tended to be) higher in participants with high expression. Finally, participants with high harm-avoidance expression were distinguished from those with medium levels by higher relative abundance of *Lactobacillus* (*p* = 0.05). 

### 3.7. Anthropometric and Biochemical Parameters

Within our participants, BMI tended to correlate positively with *Collinsella* (tau = 0.271, *p* = 0.1, Kendall rank correlation test), but negatively with *Lachnospira* (tau = −0.454, *p* = 0.01) (Figure 4C). Negative correlations were also found between *Akkermansia* and dopamine (tau = −0.449, *p* = 0.02), and between *Phascolarctobacterium* and serotonin (tau = −0.497, *p* = 0.006). 

All participants showed low calprotectin levels (<100 μg/g), except one who reported a concentration of 234 μg/g and another of 155 μg/g. Regarding cortisol, four participants reported a concentration above 19.5 μg/dL. No significant correlations were found between microbiota composition and diversity. 

## 4. Discussion

In this exploratory pilot study, we aimed to provide the first evidence about the relationship between gut microbiota and behavioral responses to fearful stimuli in women with obesity. 

First, we confirmed in our sample the presence of difficulties in detecting fearful facial expressions, in agreement with a previous study in which the same behavioral task was used [20], and in line with the main hypothesis of altered fear-processing in obesity [21,22,23,24]. Crucially, based on the results of the psychological assessment, we suggest that such an alteration is not driven by symptoms of anxiety and depression, which were not recorded in our sample. Regarding temperament, our participants showed higher expressions of harm avoidance and reward dependence, in agreement with previous evidence [27,28,29]. Finally, gut microbiota dysbiosis typically observed in obesity [30,31,32] was also confirmed in our sample.

When looking for relationships between gut microbiota profiles and sensitivity to fearful stimuli, we observed that dysbiosis was overall more severe in those participants with an altered performance (i.e., with higher reaction time and lower level of accuracy) in recognizing fearful expressions. Indeed, they were characterized by increased proportions of *Coriobacteriaceae,* especially *Collinsella*, a well-known microbial signature of obesity and metabolic disorders [47,70,71]. As far as we know, this is the first time that associations between *Collinsella* and behavioral responses to fearful stimuli have been reported. Although extreme caution must be taken in interpreting these data, given their correlative nature, it should be noted that *Collinsella* is recognized as a pathobiont that can induce the expression of interleukin 17A which, in turn, has been shown to influence neuronal functions, even independently of its pro-inflammatory activity, and to directly modulate fear behavior in mice [74]. 

Furthermore, women with poorer behavioral performance in the implicit-facial-emotion-recognition task, showed reduced proportions of a number of beneficial microbes, already inversely associated with obesity, such as: (i) the SCFA producers *Lachnospira* and *Coprococcus*; (ii) the next-generation probiotic candidate *Akkermansia*, proposed for the treatment of obesity and related complications [73], and recently approved by the European Food Safety Authority in pasteurized form [75]; and (iii) *Parabacteroides*, with known anti-obesity effects [76].

Focusing on temperament, we observed that higher expression of harm-avoidance temperament, which is related to anxiety-like symptoms, was linked to increased levels of *Lactobacillus*. This finding seems to contradict some previous evidence, which has shown that lactobacilli can attenuate anxiety- and depression-related behaviors [77]. However, it has been suggested that some *Lactobacillus* species aggravate, rather than alleviate, certain disease states, including obesity [32,78]. We also observed that participants with lower expression of persistence temperamental trait—suggesting compulsiveness and negative emotionality, especially in stressful situations [79,80]—showed elevated proportions of *Collinsella* and low amounts of health-associated taxa such as *Christensenellaceae* and *Lachnospira.* As far as we know, this is the first evidence of a relationship between *Collinsella* and temperamental traits. Finally, we observed that participants with a high expression of reward dependence temperamental trait—thus being particularly sensitive to rewards, and showing a tendency to rewarding behavior [15]—differed in microbiota composition from those with medium levels. Interestingly, it has previously been reported that individuals with high reward-sensitivity and low self-regulation respond particularly to food-associated stimuli, and have a propensity for uncontrolled eating behavior, resulting in long-lasting positive energy balance and, thereby, obesity [27].

Unfortunately, the reduced evidence in the field—especially when grounded in a behavioral approach, and not exclusively in a self-description approach (i.e., questionnaires), as in our study—may have limited the interpretation of our data. Moreover, several limitations of the study should be underlined, as well as future research directions. First, as done elsewhere [17,81], we performed a correlational study; therefore, we could not report or discuss any causal mechanisms between fear processing and microbiota composition in obesity. Another relevant limitation was that, although only a small number of participants were enrolled for this study, the experimental approach was time- and cost-consuming. Due to the small sample size, our results may have been prone to type II errors, and we may have overinterpreted or misinterpreted our data. Furthermore, only females were enrolled, and they were sampled only once, the day after the experimental task. As for gender, due to the evidence on its role in the microbiota—gut–brain axis [82,83,84] and emotional recognition (e.g., [44]), it might be useful to replicate the present study, focusing on males’ behavior. In the context of gender-specific medicine, it would be interesting to evaluate whether the interaction between psychological behavior and gut microbiota, described in our women with obesity, may be linked to circulating estrogens [85]. Other fear-related behaviors should also be investigated, such as fear-conditioning and extinction-processing, which have become an exemplary translational model for understanding and treating anxiety disorders in humans and animal models [86]. Nevertheless, more complex psychological behavior should be investigated. For example, considering that facial emotion recognition promotes communication, empathy, and social cognition [87,88], it may influence and mediate social interactions. There is multiple evidence of interpersonal difficulties in obesity [89]. Moreover, some authors have suggested an interesting and complex relationship between microbiota and sociability, which may account for social disorders in humans [90,91]. Based on these two observations, the role of microbiota alteration in social cognition, especially in the context of obesity, could be investigated in the future. Among other limitations, our participants were recruited before and after the first wave of the COVID-19 pandemic in Italy. In the literature, there is a large amount of evidence suggesting higher levels of psychological distress in the population during the COVID-19 pandemic [92,93], which in turn may have affected fear-processing. However, to our knowledge, only two studies have addressed the post-effect of COVID-19 pandemic-related distress on facial emotion recognition abilities. The first study was conducted during the first Italian lockdown (from 12 April to 3 May 2020). Scarpina and colleagues [94] used a short version of the task used in the present article, investigating possible alterations in the recognition of the facial expression of fear in the Italian population, through an online experiment. Crucially, the general population showed a preserved ability to recognize fearful expressions, even though they reported higher levels of psychological distress. The second study was conducted by Antico and Corradi Dell’Acqua [95] during the lockdown between 7 September and 19 November 2020, in Switzerland. The authors verified whether either form of social segregation influenced the processing of pain, disgust, or neutral facial expressions. The lockdown negatively affected the processing of pain-specific information, without influencing other components of the affective facial response, related to disgust or broad unpleasantness. However, in the present study, we did not test any facial expression of pain, which may have been more affected by the lockdown. Therefore, taking into account this previous evidence, we suggest that the behavioral performance shown by our participants may not have been related to the COVID-19 pandemic-related distress. As regards gut microbiota profiling, we used a gold-standard technique (*i.e.*, 16S rRNA amplicon sequencing), which did not, however, allow a fine resolution of the compositional structure (including components other than the bacterial one) nor the recovery of functional information. Finally, the enrollment of an *ad-hoc* control (normal-weight) group was not foreseen, but we used normative data for psychological functioning, and previously published sequence data for the gut microbiota. However, it must be said that comparison with already-existing data is a well-established approach in the literature, in both psychological [96,97] and gut microbiota [47,71] research fields. Furthermore, the behavioral difference between a sample of women affected by obesity and healthy normal-weight women (i.e. the control group), in regard to the implicit-facial-emotion-recognition task, had already been reported [20]. Future studies necessarily require replications in larger cohorts, possibly with multi-omics approaches, to deepen our knowledge of microbiota–brain interactions, with particular regard to behavioral performance and temperamental traits. In an effort to move beyond associative observations, potential mechanistic insights should be validated in an animal model. In such studies, an *ad-hoc* control group to be compared with the experimental group should also be recruited to carry out the experimental task.

## 5. Conclusions

In addition to confirming that women affected by obesity may show difficulties in recognizing fearful expressions [20] and may have unbalanced gut microbiota [30,31,32,33], this exploratory pilot study suggested that altered behavioral fear-related performance, and the expression of specific temperamental traits, could be linked to distinct microbiota components. Although preliminary, this study may represent the first step in increasing our knowledge of the role of the gut–brain axis in human fear-related processing. Also, it could pave the way for new rehabilitative treatments, in which diets aimed at modulating the microbiota, and psychological intervention, could accelerate a higher quality of life in individuals with obesity.

## Figures and Tables

**Figure 1 nutrients-14-03788-f001:**
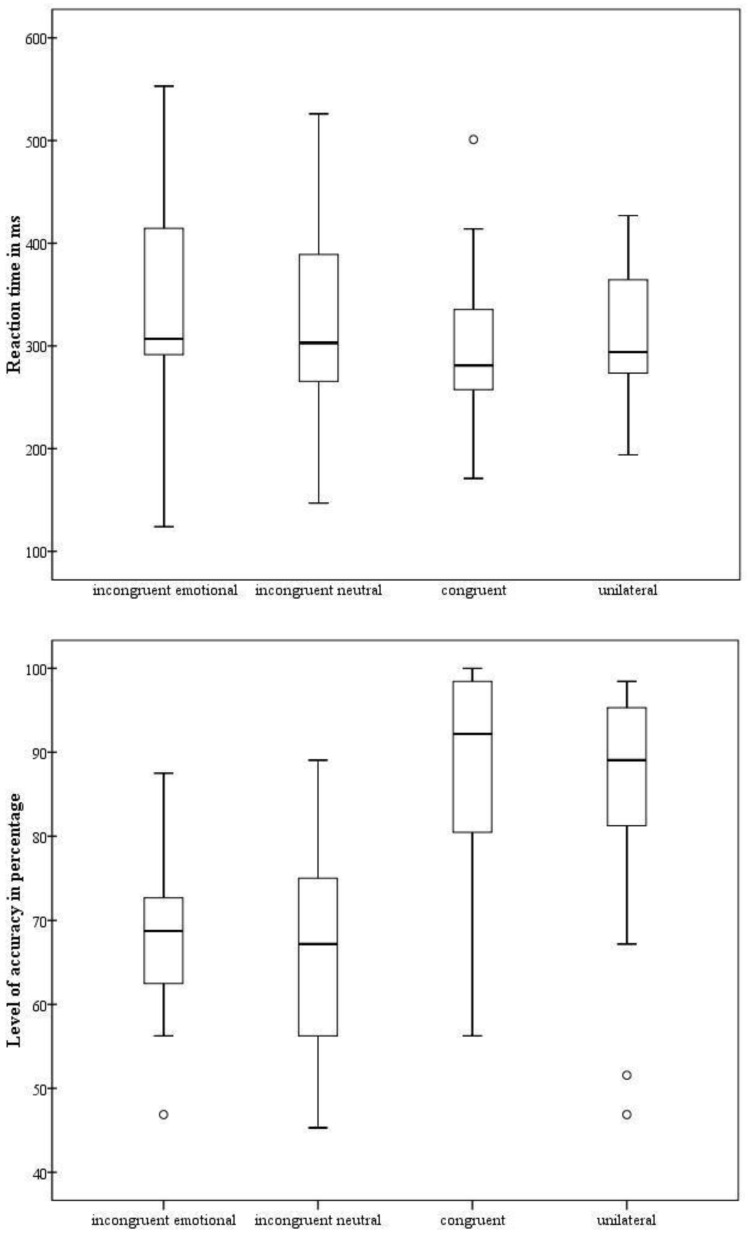
Behavioral results of the experimental task. For each experimental condition (bilateral incongruent emotional, bilateral incongruent neutral, bilateral congruent, and unilateral), the reaction time in milliseconds (**upper panel**), and the level of accuracy in percentage (**lower panel**), are shown. The minimum, lower quartile, median, upper quartile, maximum, and outliers are displayed.

**Figure 2 nutrients-14-03788-f002:**
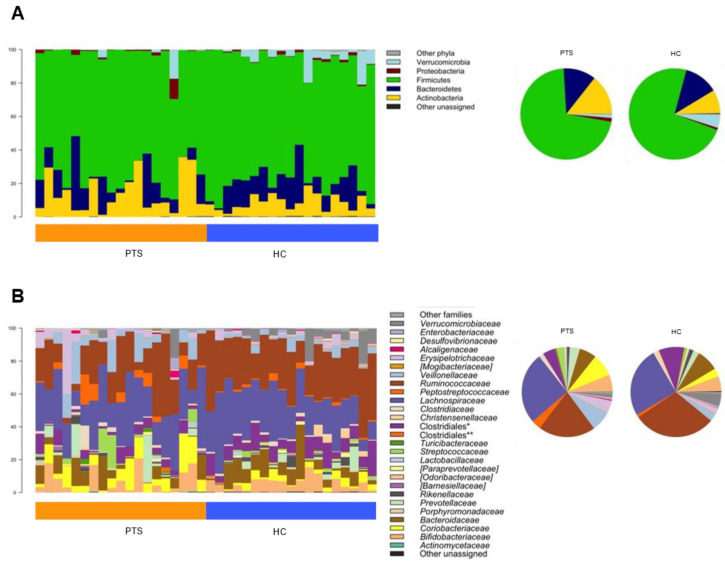
Phylum- and family-level composition of the gut microbiota of women affected by obesity, compared to normal-weight women. Relative abundance profiles of the gut microbiota of women affected by obesity (PTS) compared to age-matched, normal-weight women from the same geographical area (HC), at phylum (**A**) and family (**B**) level. For each panel, left: bar graphs of the individual profiles; right: pie charts showing average values.

**Figure 3 nutrients-14-03788-f003:**
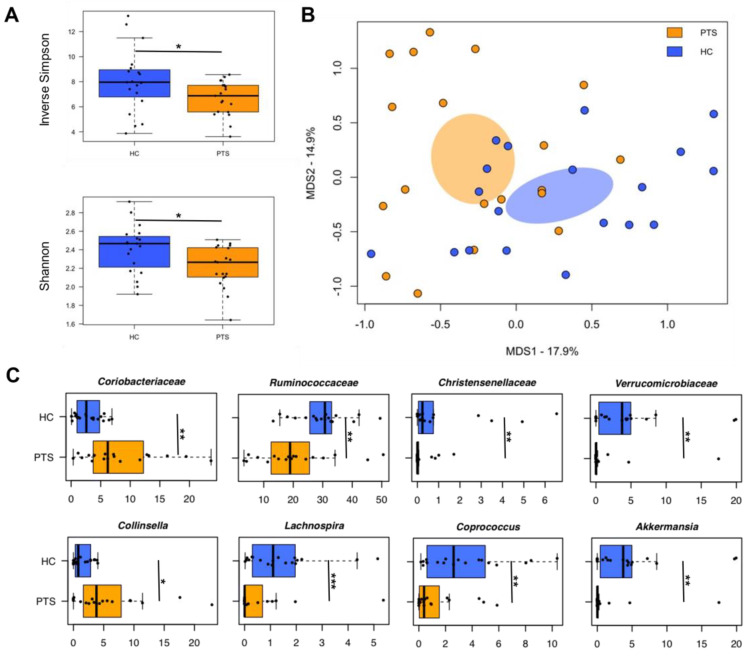
Diversity and taxonomic signatures of the gut microbiota of women affected by obesity, compared to normal-weight women. (**A**) Boxplots showing the distribution of alpha diversity, according to the inverse Simpson and Shannon index, in women affected by obesity (PTS), compared to age-matched, normal-weight women from the same geographical area (HC). A lower value was found in participants (*p* ≤ 0.04, Wilcoxon test). (**B**) PCoA plot of beta diversity, based on Bray–Curtis dissimilarity between the genus-level profiles. A separation between groups was found (*p* = 0.0005, PERMANOVA). Samples were identified with colored dots, as in A. Ellipses include 95% confidence area based on the standard error of the weighted average of the sample coordinates. (**C**) Boxplots showing the relative abundance distribution of differentially represented taxa between groups. * *p* ≤ 0.05, ** *p* ≤ 0.01, *** *p* ≤ 0.001, Wilcoxon test.

**Figure 4 nutrients-14-03788-f004:**
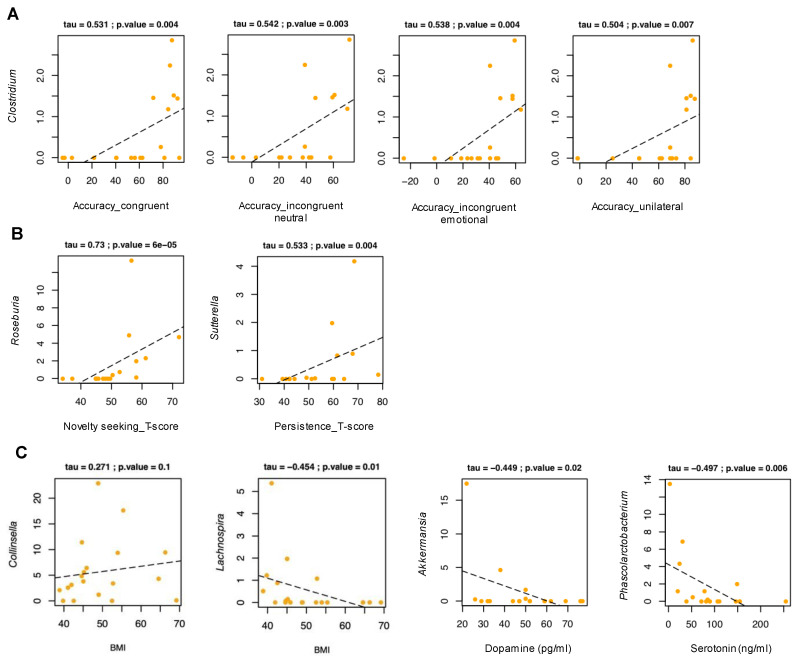
Scatter plots of correlation between relative taxon abundances and experimental task parameters, temperamental traits, and anthropometric and biochemical variables in women affected by obesity. Only statistically significant correlations (*p* ≤ 0.05), based on the Kendall rank correlation test, are shown for experimental task parameters (**A**), temperamental traits (**B**), and anthropometric and biochemical variables (**C**). BMI, body mass index.

**Figure 5 nutrients-14-03788-f005:**
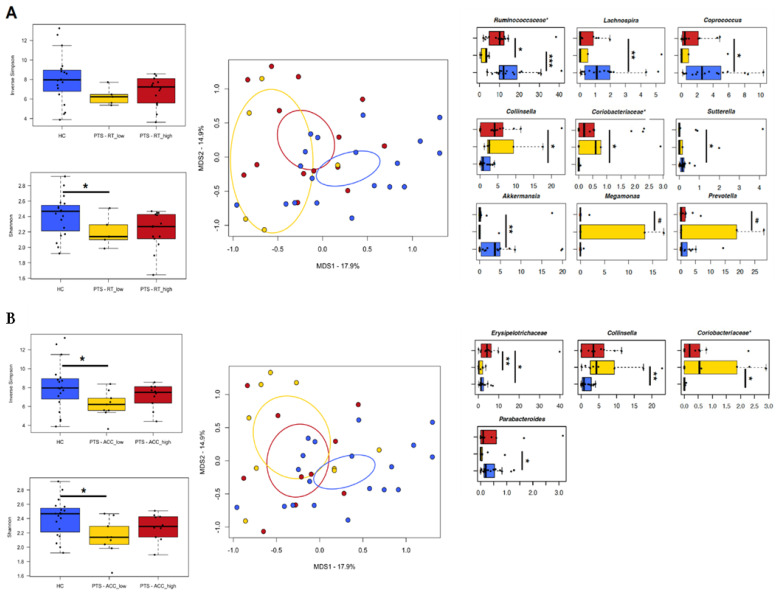
Variation of the gut microbiota of women affected by obesity in relation to experimental task parameters. Study participants (PTS) were stratified by high or low value of each behavioral parameter ((**A**), reaction time (RT); (**B**), level of accuracy (ACC)), and compared to age-matched, normal-weight women (HC). For each panel, left: boxplots showing the distribution of alpha diversity, according to the inverse Simpson and Shannon index, in the study groups; center: PCoA plot of beta diversity, based on the Bray–Curtis dissimilarity between the genus-level profiles, with ellipses including 95% confidence area based on the standard error of the weighted average of sample coordinates; right: boxplots showing the relative abundance distribution of differentially represented taxa between groups. An asterisk next to the family name indicates unclassified genera. * *p* ≤ 0.05, ** *p* ≤ 0.01, *** *p* ≤ 0.001, ^#^ *p* ≤ 0.1, Wilcoxon test.

**Figure 6 nutrients-14-03788-f006:**
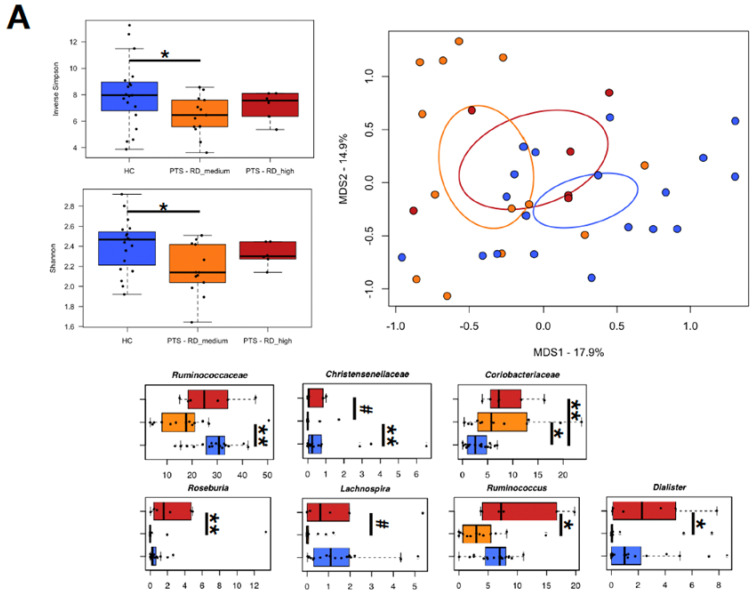
Variation of the gut microbiota of women affected by obesity in relation to temperamental traits. Study participants (PTS) were stratified by T-scores in high, medium or low expression for each of the temperamental traits ((**A**), reward dependence (RD); (**B**), persistence (P); (**C**), harm avoidance (HA); (**D**), novelty-seeking (NS)), and compared to age-matched, normal-weight women from the same geographical area (HC). For each panel, top left: boxplots showing the distribution of alpha diversity, according to the inverse Simpson and Shannon index, in the study groups; top right: PCoA plot of beta diversity, based on Bray–Curtis dissimilarity between the genus-level profiles, with ellipses including 95% confidence area based on the standard error of the weighted average of sample coordinates; bottom: boxplots showing the relative abundance distribution of differentially represented taxa between groups. * *p* ≤ 0.05, ** *p* ≤ 0.01, ^#^ *p* ≤ 0.1, Wilcoxon test.

**Table 1 nutrients-14-03788-t001:** Clinical, psychological, and biochemical characteristics of study participants.

Parameter	Mean	Standard Deviation	Range
Age (years)	42.78	6.11	27–53
Education (years)	14	3.5	8–18
Body mass index (kg/m^2^)	49.58	8.98	38.85–69.18
Waist circumference (cm)	126.98	17.49	100.5–170
Fat-free mass (%)	42.59	5.56	31.6–49.1
Fat mass (%)	57	5.72	48.5–68.4
Systolic pressure (mmHg)	135.25	12.51	120–160
Diastolic pressure (mmHg)	83	8.8	70–100
Heart rate (beats per minute)	86.6	11.44	63–114
Total cholesterol (mg/dL)	186.7	33.02	120–247
HDL cholesterol (mg/dL)	47.05	7.34	39–74
LDL cholesterol (mg/dL)	122.00	29.11	66–177
Triglycerides (mg/dL)	131.9	41.76	55–207
C-Reactive protein (mg/dL)	1.52	1.29	0.1–4.8
AST (mg/dL)	24.55	15.32	13–67
ALT (mg/dL)	27.9	21.22	9–88
Insulin (µU/mL)	18.52	11.22	6.2–41.8
Fasting glucose (mg/dL)	102.50	13.62	87–129
HbA1c (%)	5.94	0.82	4.9–8.9
**Psychological General Wellbeing Index**			
Anxiety (min-max = 0–25)	15.84	4.48	4–21
Depression (min-max = 0–15)	11.94	1.74	8–14
Positive wellbeing (min-max = 0–20)	9.89	2.8	6–15
Self-control (min-max = 0–15)	10.36	2.69	6–15
General health (min-max = 0–15)	8.42	3.35	3–15
Vitality (min-max = 0–20)	10.15	3.33	1–16
Total score (min-max = 0–100)	66.63	13.22	43–94
**Biochemical components**			
Cortisol (μg/dL)	14.78	5.93	7.7–27.5
Adrenaline (pg/mL)	358.5	214.88	130–99
Noradrenaline (pg/mL)	79.6	29.97	36–138
Dopamine (pg/mL)	50.3	17.34	22–82
Serotonin (ng/mL)	95.32	60.97	3.06–254.6
N = 19.			

**Table 2 nutrients-14-03788-t002:** Response percentage (%) for each item of the Medi-lite questionnaire for study participants.

	Consumption Frequency		
fruit	<1 s/d	1–2 s/d	>2 s/d
	46.6%	40.0%	13.3%
vegetables	<1 s/d	1–2 s/d	>2 s/d
	26.66%	66.7%	6.7%
pulses	<1 s/w	1–2 s/w	>2 s/w
	60.0%	33.3%	6.7%
cereals	<1 s/d	1–1.5 s/d	>1.5 s/d
	0%	13.3%	86.7%
fish	<1 s/w	1–2.5 s/w	>2.5 s/w
	40.0%	53.3%	6.7%
meat and cured meats	<1 s/w	1–1.5 s/w	>1.5 s/w
	0%	66.7%	33.3%
dairy products	<1 s/d	1–1.5 s/d	>1.5 s/d
	13.3%	66.7%	20.0%
alcohol	<1 s/d	1–2 s/d	>2 s/d
	100%	0%	0%
olive oil	occasionally	frequently	regularly
	26.7%	33.3%	40.0%

N = 19; s/d = serving/day; s/w = serving/week.

**Table 3 nutrients-14-03788-t003:** Mean, standard deviation, and range for each score at the Temperament and Character Inventory for studied participants. For each temperamental trait, we indicate the minimum (min) and maximum (max) score. Significant differences with the normative sample [56] are shown in bold (*p* < 0.05).

	Mean	Standard Deviation	Range	Statistical Results
Novelty-Seeking (min–max = 0–175)	98.89	11.28	78–127	t = 0.13; *p* = 0.89; 95% CI (−5.48; 6.26)
Harm Avoidance (min–max = 0–165)	105.26	16.65	72–135	t = 2.63; ***p* = 0.008;** 95% CI (2.26; 15.45)
Reward Dependence (min–max = 0–150)	108.36	11.96	85–127	t = 2.3; ***p* = 0.02;** 95% CI (1.03; 12.88)
Persistence (min–max = 0–175)	119.84	18.01	89–157	t = 0.96; *p* = 0.33; 95% CI (−3.37; 9.85)

N = 19; df = 757; CI = confidence interval.

## Data Availability

Raw sequencing reads are available in the National Center for Biotechnology Information Sequence Read Archive (BioProject ID: PRJNA837468).

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
