# Peer review of "Gut Microbiota and Fear Processing in Women Affected by Obesity: An Exploratory Pilot Study"

_nutrients, 2022, doi:10.3390/nu14183788_

Round 1

Reviewer 1 Report

In the manuscript “Gut microbiota and fear processing in women affected by obesity: an exploratory pilot study by Scarpina et al. An exploratory pilot study explaining role gut-brain axis with fear in obesity. I have following comments,  

1.     In introduction rational for selecting female is not given

2.     Limitations of study should be mentioned.

3.     Normal lean group should be included to compare obese group.

4.  Authors have shown body mass index (39-69), is there any rational to selection?

5.     Please correct order of range for SBP, write 120-160 not 160-120.

6.     Figure 1 is illegible.

7.     In table 1, Number of participants needs to be included.

8.     In table mean BMI is 49.14 (assuming N=20), while in line 381 it is 40.1 for N=19. Please provide proper calculation and justification.

Author Response

Reviewer #1

In the manuscript “Gut microbiota and fear processing in women affected by obesity: an exploratory pilot study by Scarpina et al. An exploratory pilot study explaining role gut-brain axis with fear in obesity. I have following comments,  

  1. In introduction rational for selecting female is not given

REPLY. We thank the Reviewer for this comment and apologize for not specifying why we only selected females in the previous version of the manuscript. Following this suggestion, we have revised the Introduction section, making it clear that we enrolled females for different methodological reasons (lines 114-125 and also lines 140-141 of the Methods section). First, gender-specific differences in body morphology and particularly fat distribution in obesity are well recognized (e.g., Blaak, 2001; Haslam, 2005; Kanter and Caballero, 2012). Moreover, gut microbiota has been reported to vary by gender, and estrogen and androgen metabolism appears to be related to the gut microbiota profile (for a recent review, please see Yoon and Kim, 2021). Crucially for the topic of this work, females and males differ in emotional experience and expression (e.g., Brody and Hall, 1993; Abbruzzese et al., 2019). Finally, we only enrolled women with obesity for reasons of comparability of the experimental data shown in this work with the evidence reported in our previous study (Scarpina et al., 2021c), in which we tested a group of women affected by obesity on the same emotional task. Of course, as specified in the paragraph on the limitations of this study, reported at the end of the Discussion (lines 651-658), the results from this study on women’s performance should be very carefully extended to males.

  1. Limitations of study should be mentioned.

REPLY. We thank the Reviewer again for her/his comment. In the revised version of our manuscript, we have implemented the last paragraph of the Discussion, by clarifying the main limitations of our study, which are briefly summarized below: small sample size, absence of an ad-hoc control group, recruitment of women only, recruitment carried out before and after the first wave of the COVID-19 pandemic in Italy, single-point sampling, use of a compositional (and non-functional) profiling technique for the gut microbiota, exploratory and correlative nature of the study (and therefore lack of mechanistic insights), and non-assessment of other fear-related or more complex psychological behaviors. In the same paragraph, we specified how these limitations could be overcome in future research. Please, see lines 642-708.

We would also welcome any other possible limitations to our study recognized by the Reviewer.

  1. Normal lean group should be included to compare obese group.

REPLY. We strongly agree with the Reviewer on the added value of a control (normal-weight) group recruited ad hoc for the present study. However, the main goal of our pilot study was to test the relationship between the ability to process fearful stimuli and the composition/diversity of the gut microbiota within a sample of women affected by obesity, i.e., to conduct a correlational study, as done elsewhere in infant cohorts (Xie et al., 2022 Aatsinki et al., 2020). Therefore, in light of this goal, the enrollment of a control group was not foreseen. On the other hand, the behavioral difference between a sample of women affected by obesity and healthy normal-weight women at the implicit facial emotion recognition task had already been reported in a previous published study (Scarpina et al., 2021c) (see also the Introduction section, lines 122-125, and the Discussion section, lines 700-702): in particular, this study has demonstrated that women affected by obesity suffer from an altered processing of fear stimuli, in line with other studies in the field (for a review, please refer to Scarpina and Vaioli, 2021).

Regarding the gut microbiota, it must be said that it is common practice in microbiota studies to include already published data from previous studies for comparative purposes (see for example Peled et al., 2020). Clearly it is desirable to recover data from subjects matched as accurately as possible to study participants for major microbiota-associated confounding factors (e.g., age, gender and geography) (Vujkovic-Cvijin et al., 2020) and whose samples have been processed following the same wet and in-silico procedures, as was done in our work. In the revised version of our manuscript, we have made this point clearer (see lines 298-307, 697-700).

Nevertheless, in the final paragraph of the Discussion, where we discussed the main limitations of our study, we also suggested replicating our findings by adopting a “two-group experimental design” (see lines 694-702, 707-708). However, the opportunity to use such a methodological design in the clinical setting (as in our case) must always be carefully considered also from an economic point of view.

  1. Authors have shown body mass index (39-69), is there any rational to selection?

REPLY. We thank the Reviewer for this comment: indeed, in the first version of the manuscript we did not report what was the rationale for selecting our participants with obesity. We apologize for this lack. As specified in the new version of the manuscript (Methods section, lines 143-144), according to WHO (2006), obesity in adults (men and women) is defined as a body mass index of 30 kg/m2 or higher. Therefore, we selected participants meeting this criterion.

  1. Please correct order of range for SBP, write 120-160 not 160-120.

REPLY. We apologize for the inaccuracy. In the revised version of our manuscript (Table 1), we corrected the range for SBP.

  1. Figure 1 is illegible.

REPLY. We apologize for the inconvenience. In the revised version of our manuscript, we have included a more readable version of Figure 1.

  1. In table 1, Number of participants needs to be included.

REPLY. The Reviewer is right, and we apologize for the inaccuracy. Following this comment, for each table, we have specified the number of participants (see also the answer to the next point).

  1. In table mean BMI is 49.14 (assuming N=20), while in line 381 it is 40.1 for N=19. Please provide proper calculation and justification.

REPLY. We apologize for the mistake we made in reporting the data. Also in light of Reviewer #2’s comments, we reviewed all the tables, figures and data provided in the text, always referring to 19 (instead of 20) study participants. As specified in the manuscript, we initially did a convenience sampling of 20 participants but, as one fecal sample was missing, the gut microbiota data are shown for only 19 subjects. At this point, for consistency reasons, we have chosen to present all data for the same number of participants. Please, see the new Figure 1, Tables 1-3 and lines 337-340, 371, 375-378, 383, 385-386.

Reviewer 2 Report

The graphics are of poor resolution. The text is often not legible. Especially in the compound figures.

The authors hypothesize that diet-related and thus intestinal flora affect fear perception. The authors have already pre-selected only right-handed obese women in an attempt to reduce the number of variables. Why specifically? That raises the question of what other factors influence fear perception? These variables are then not accounted for. Patients with diagnosed psychological disordered were removed, but cases of mild undiagnosed social anxiety are still possibly present. My main criticism therefore is that a correlation might exist, but no provable causal mechanism is described. These results and conclusions are therefore pure conjecture and are easily over-interpreted. A number of other factors might conceivably explain the same results.

Comparison with data from a different data-set where the testing criteria and selection criteria for participants is not known or different is risky, and speculative, at best.

The time between fecal sample collection of the temperament and facial recognition task should be clearly indicated. What was the period between the tests? Was it constant? Can the fecal data, therefore, be reliably compared to the facial recognition data? And therefore also to the temperament data? Or where all data collected on the same day?

Why was fecal material collected from only 19 women? The methods section clearly states that 20 women were recruited. If only partial data exists for 19 – why was the extra person still included in the facial recognition and temperament data? 

Author Response

Reviewer #2

The graphics are of poor resolution. The text is often not legible. Especially in the compound figures.

REPLY. We thank the Reviewer for this comment and apologize for any inconvenience. We have completely revised the figures in our manuscript, in order to improve their resolution and readability.

The authors hypothesize that diet-related and thus intestinal flora affect fear perception. The authors have already pre-selected only right-handed obese women in an attempt to reduce the number of variables. Why specifically? That raises the question of what other factors influence fear perception? These variables are then not accounted for. Patients with diagnosed psychological disordered were removed, but cases of mild undiagnosed social anxiety are still possibly present. My main criticism therefore is that a correlation might exist, but no provable causal mechanism is described. These results and conclusions are therefore pure conjecture and are easily over-interpreted. A number of other factors might conceivably explain the same results.

REPLY. We are very grateful to the Reviewer for raising this point, which helped us revise our manuscript in order to avoid overstatements in the interpretation of our data and any references to causal mechanisms. Indeed, since the aim of the present pilot study was to test the relationship between the ability to process fearful stimuli and the composition/diversity of the gut microbiota within a sample of women affected by obesity, i.e., to conduct a correlational study as done elsewhere (Xie et al., 2022; Aatsinki et al., 2020), it is obviously not possible to describe any causal mechanism. We have specified these aspects (i.e., the absence of mechanistic insights and the exploratory and correlative nature of our study) in the final paragraph of the Discussion, where we reported and discussed the limitations of our study, as well as in the Conclusions section (please, see lines 645-648, 705-707, 715 and also the answer to the second point raised by Reviewer #1). Furthermore, we have toned down any overstatements throughout the main text (lines 611-613, 648-651, 691-694, 702-705, 707-708, 710-716).

In this comment, the Reviewer raised another interesting, albeit very difficult to be solved, issue. Indeed, obesity is a multifactorial clinical condition, the components of which may variably impact physical and psychological health. Because of that, stringent criteria are needed in enrolling, as we have tried to do in this work. In particular, in the Introduction section, we have now provided the reasons why we only enrolled females (lines 114-125, and also lines 140-141 of the Methods section): i) gender-specific differences in body morphology and particularly fat distribution in obesity are well recognized (e.g., Blaak, 2001; Haslam, 2005; Kanter and Caballero, 2012); ii) gut microbiota has been reported to vary by gender, and estrogen and androgen metabolism appears to be related to the gut microbiota profile (for a recent review, please see Yoon and Kim, 2021); iii) females and males differ in emotional experience and expression (e.g., Brody and Hall, 1993; Abbruzzese et al., 2019); and iv) we wanted to compare the experimental data obtained in this work with the evidence reported in our previous study (Scarpina et al., 2021c), in which we tested a group of women affected by obesity on the same emotional task.

As for the physical (health-related) criteria, we used the same ones reported by Cancello and colleagues (2019), who profiled the gut microbiota in a sample of individuals with obesity. Specifically, these criteria are in line with the definition of “metabolically healthy obesity” (for a review, please see Blüher, 2020), defined by the absence of any metabolic disorders and cardiovascular disease, including type 2 diabetes, dyslipidemia, hypertension, and atherosclerotic cardiovascular disease (ASCVD) in a person with obesity. For clarity, we have now specified this point in the Methods section (lines 151-152).

As correctly stated by the Reviewer, it must also be considered that multiple psychological factors may affect not only the facial emotion recognition, but also fear processing. Indeed, there is a plentitude of studies in the literature on these effects, and most of the research is still ongoing. Considering that, we have adopted a very conservative approach, using the same constraints in the recruitment reported in Scarpina and colleagues (2019, 2021c). For example, we only recruited women, for the reasons previously reported. About handness, in experimental studies, only right handers are generally recruited (excluding left handers) when a motor response (as in our case) is required by the task. Again, we have specified this point in the Methods section (lines 140-143). With specific regard to the psychological components, we asked our participants to fill out the Italian version (Grossi et al., 2006) of the Psychological General Well Being Index (PGWBI) (Dupuy, 1984). This well-known clinical questionnaire is used to describe the subjective perception of well-being, according to six dimensions: anxiety, depressed mood, positive well-being, self-control, general health, and vitality. Crucially for the aim of our exploratory pilot study, there were no differences in scores related to anxiety and depression, parameters that might impact the facial emotion recognition task (Kang et al., 2019; Krause et al., 2021). In her/his comment, the Reviewer mentioned social anxiety, which implicitly recalls the concept of social functioning. It would be very interesting to verify whether microbiota alterations in obesity (but also in healthy and other clinical conditions) may affect social functioning, as proposed by some papers (Parashar and Udayabanu, 2016; Sherwin et al., 2019). As we know, no previous study has covered this research field. Thus, in the last part of the Discussion, we have introduced a new sentence in which we have stressed the importance of exploring this field in the future (lines 662-670). Nevertheless, we would underline that according to the DSM-V, the criteria for a diagnosis of social anxiety disorder include: “persistent, intense fear or anxiety about specific social situations because you believe you may be judged negatively, embarrassed or humiliated. Avoidance of anxiety-producing social situations or enduring them with intense fear or anxiety”. So, even though social anxiety grounds phenomenologically on the emotion of fear, in the present study we tested a different psychological mechanism, which is the unintentional and unconscious visual process of facial emotion recognition (Ekman, 1992; Vuilleumier et al., 2002).

Finally, the Reviewer stated that “a number of other factors might conceivably explain the same results”, but unfortunately they were not explicitly reported. Even though in principle we may agree with the Reviewer’s comment, on the other hand not all the (at least known) factors may be introduced as controlled variables in a single study, especially when a research topic is still unexplored. In any case, if the Reviewer could kindly suggest to us which other factors she/he refers to, we will be more than happy to report and discuss them in our manuscript. We thank the Reviewer in advance.

Comparison with data from a different data-set where the testing criteria and selection criteria for participants is not known or different is risky, and speculative, at best.

REPLY. Again, we thank the Reviewer for raising this point and apologize for not addressing it properly in the previous version of our manuscript.

The main aim of our pilot study was to investigate the relationship between the fear processing/temperamental traits and the composition/diversity of the gut microbiota within a sample of women with obesity. Therefore, in light of this goal, the enrollment of a control (normal-weight) group was not foreseen.

However, to describe the psychological functioning of our participants, we used standardized psychological questionnaires, in other words questionnaires about which normative data are available in the literature. Indeed, when assessing the psychological functioning (or other clinical parameters) of a single individual or a cohort, clinical normative data are particularly useful because they allow clinicians and researchers to describe their participants’ performance in terms of how they are functioning in comparison to a larger population. Several examples of this very traditional approach can be found in the literature. For more details, the following articles can be consulted:

  • Kendall, P. C., & Sheldrick, R. C. (2000). Normative data for normative comparisons. Journal of consulting and clinical psychology, 68(5), 767.
  • Mitrushina, M., Boone, K. B., Razani, J., & D'Elia, L. F. (2005). Handbook of normative data for neuropsychological assessment. Oxford University Press.

Thus, as suggested in these papers and as traditionally done in psychological research, we compared our participants’ scores at the psychological questionnaires with the normative data related to the Italian population reported in the seminal articles. Specifically, we considered the normative data reported by Grossi and colleagues (2006) for PGWBI, and the normative data published by Martinotti et al. (2008) for TCI (Temperament and Character Inventory).

Regarding the gut microbiota, it must be said that it is common practice in microbiota studies to include already published data from previous studies for comparative purposes (see for example Peled et al., 2020). Clearly it is desirable to recover data from subjects matched as accurately as possible to study participants for major microbiota-associated confounding factors (e.g., age, gender and geography) (Vujkovic-Cvijin et al., 2020) and whose samples have been processed following the same wet and in-silico procedures, as was done in our work. In the revised version of our manuscript, we have made this point clearer (see lines 305-307, 694-700).

That said, in the final paragraph of the Discussion, where we discussed the main limitations of our study, we suggested replicating our findings by adopting a “two-group experimental design” (see lines 707-708). However, the opportunity to use such a methodological design in the clinical setting (as in our case) must always be carefully considered also from an economic point of view.

The time between fecal sample collection of the temperament and facial recognition task should be clearly indicated. What was the period between the tests? Was it constant? Can the fecal data, therefore, be reliably compared to the facial recognition data? And therefore also to the temperament data? Or where all data collected on the same day?

REPLY. We thank the Reviewer for this comment. In the revised version of our manuscript, we have specified that fecal and blood samples were always collected the day after the experimental task and psychological assessment (lines 219-220, 222-223). This procedure was consistently used for all participants.

Why was fecal material collected from only 19 women? The methods section clearly states that 20 women were recruited. If only partial data exists for 19 – why was the extra person still included in the facial recognition and temperament data? 

REPLY. The Reviewer is right and we apologize for the inaccuracy. As noted, one fecal sample was missing, so the gut microbiota data are shown for 19 subjects, while the psychological ones (specifically, in the experimental task and temperament) for the 20 subjects initially enrolled. In the revised version of the manuscript, for consistency reasons, we reviewed all the tables, figures and data provided, always referring to 19 (instead of 20) study participants. Please, see the new Figure 1, Tables 1-3 and lines 337-340, 371, 375-378, 383, 385-386.  

Round 2

Reviewer 1 Report

I have no further comments.

Reviewer 2 Report

Well done!